# Output-Only Time-Varying Modal Parameter Identification Method Based on the TARMAX Model for the Milling of a Thin-Walled Workpiece

**DOI:** 10.3390/mi13101581

**Published:** 2022-09-22

**Authors:** Junjin Ma, Xinhong Yan, Yunfei Li, Haoming Li, Yujie Li, Xiaoyan Pang

**Affiliations:** 1School of Mechanical and Power Engineering, Henan Polytechnic University, Jiaozuo 454000, China; 2College of Computer Science and Technology, Henan Polytechnic University, Jiaozuo 454000, China

**Keywords:** milling, machining dynamics, modal identification, thin-walled parts, natural frequency

## Abstract

The process parameters chosen for high-performance machining in the milling of a thin-walled workpiece are determined by a stability prediction model, which needs accurate modal parameters of the machining system. However, the in-process modal parameters are different from the offline modal parameters and are difficult to precisely obtain due to material removal. To address this problem, an accurate time-dependent autoregressive moving average with an exogenous input (TARMAX) method is proposed for the identification of the modal parameters in the milling of a thin-walled workpiece. In this process, a TARMAX model considering external force excitation is constructed to characterize the actual condition in the milling of a thin-walled workpiece. Then, recursive method and sliding window recursive method are used to identify TARMAX model parameters under time-varying cutting conditions. Subsequently, a three-degree of freedom (3-DOF) time-varying structure numerical model under theoretical milling forces and white-noise excitation is established, and the computational results show that the predicted natural frequencies using the proposed method are in close agreement with the simulated values. Finally, several experiments are designed and carried out to validate the effectiveness of the proposed method. The experimental results show that the predicted accuracy of the proposed method using actual cutting forces is 95.68%. Good agreement has been drawn in the numerical simulation and machining experiments. Our further research objectives will focus on the prediction of the damping ratios, modal stiffness, and modal mass.

## 1. Introduction

Thin-walled workpieces such as aero-engine blades, casings, and impellers are widely used in the aerospace industry [1] because of their light weight, high strength, and heavy load-bearing characteristics [2,3]. However, machining chatter and dimensional error caused by the problem of low rigidity occur in the milling of a thin-walled workpiece, which can affect the surface quality, machining productivity, and tool life [4,5,6]. In the milling of thin-walled workpieces, the methods of milling stability prediction are extensively investigated for avoiding chatter [7,8,9]. Generally, accurate stability lobe diagrams calculated by solving a time period delay differential equation are critical for milling stability [10,11,12]. In this process, the modal parameters of the pressing system are important factors affecting accurate stability lobe diagrams. In particular, the modal parameters are time-varying due to material removal and the tool–workpiece contact position in the milling of a thin-walled workpiece [12,13,14,15], which will lead to low cutting process parameters in milling and will reduce the machining productivity [10]. Therefore, it is necessary to accurately identify the time-varying modal parameters of thin-walled workpieces in milling in order to improve their efficiency and quality.

For extracting the modal parameters, the common method that is used is the hammer test. However, this method cannot be adopted to obtain modal parameters at in-process machining [16,17]. To deal with this problem, operational modal analysis (OMA) is used to identify the dynamic modal parameters in the milling of a thin-walled workpiece. For the efforts of identifying the modal parameters in milling, OMA can be divided into two categories: frequency domain modal identification and time domain modal identification. For frequency domain modal identification, frequency response functions or power density functions are used to obtain time-varying modal parameters, which are mainly applied in large-scale bridge and building structure health monitor fields. Nevertheless, in milling, excitation signals mainly contain strong harmonic excitation and white-noise excitation due to spindle rotation and material removal [18], which may result in the harmonic components being mistaken for structural modes in the OMA frequency domain identification method. Liang et al. [19] eliminated the response generated by harmonic excitation by the properties of white-noise excitation signals. Kiss et al. [20] employed a comb filter to eliminate the response generated by harmonic excitation. Yuan et al. [21] used Kalman filters, which effectively attenuate spindle frequency and its harmonics. However, if the harmonic fundamental frequency cannot be determined before using these methods, even a small error will be significantly amplified in the high-order harmonic components and cause filter failure. For this, Liu et al. [18] proposed a modified least square method to fit harmonics with multiple fundamental frequencies and to extract the modal parameters of the workpiece–tool system. Weijtjens et al. [22] estimated the structural dynamics of the system by using transfer functions that could be unaffected by periodic harmonic excitations. Wan et al. [23] expressed the power spectral density matrix as a spectral decomposition form with modal parameters and an extract damping ratio using inverse Fourier transform, which ignored the natural frequency identification. These methods eliminated harmonic components in machining signals and identified the modal parameters, but they relied on time domain signals to decrease the conversion errors.

For the time domain modal identification, the modal parameters could be directly identified in the time domain and avoid signal errors compared with the frequency domain method. Li et al. [24] extracted machine tool modal parameters through the stochastic subspace identification (SSI) method. Then, the effectiveness of the SSI method was validated by Yan et al. [25]. However, the identification accuracy and the time-varying tracking ability of the SSI method need to be improved. Subsequently, Burney et al. [26] employed a time-dependent autoregressive moving average (TARMA) model to estimate the machine tool modal parameters. In addition, Kim et al. [27] applied TARMA to the milling operations. Then, Zaghbani et al. [28] adopted the TARMA approach to estimate the modal parameters in real-time during machining. Ma et al. [29] proposed a kernelized TARMA model that extended on the TARMA approach, and the computational efficiency was improved. However, these methods only identified modal parameters through white-noise signals. In the actual machining, Kang et al. [30] obtained the actual modal parameters by removing the harmonic components based on the statistical characteristics of the response, but the amount of calculations were too large, and the signal produced errors due to Fourier transform. Zhuo et al. [31] adopted the TARMA method to identify the modal parameters in the milling of a thin-walled workpiece, and false modes eliminated the convergence properties of the stability lobe diagram.

Through the above analysis, it was found that a number of methods for modal parameter identification are used in a wide field of processing. However, for time-varying processing conditions, especially the milling of thin-walled workpieces, the identification of time-varying modal parameters relevant to the prediction of the system stability is less well studied. To overcome this problem, a modified TARMAX modal parameters identification method considering the actual cutting force excitation was proposed in this paper. Taking the dynamic milling force and environment noise as excitation signals, the TARMAX modal parameter identification algorithm was derived based on “frozen time”, and the relationship between time-varying autoregressive coefficients and modal parameters was established. Subsequently, the white-noise excitation signal and the time-varying autoregressive coefficients were estimated by the least square method, and then, the time-varying modal parameters were determined. This paper is organized as follows. In Section 2, the TARMAX model based on milling force excitation was established, the recursive estimation of the TARMAX model parameters was identified, and recursive estimation of the time-varying coefficient sliding windows was determined. Then, in Section 3, a 3-DOF time-varying machining system numerical simulation model was established and the modal parameter identifications were obtained and compared under different noise and window numbers. Subsequently, the effectiveness and feasibility of the proposed method were validated by several machining experiments in Section 4. Finally, some conclusions are drawn in Section 5.

## 2. TARMAX Model Modal Identification Algorithm

TARMA methods are computationally small and can capture the time-varying properties of structures in real-time. In contrast with the identification methods in the frequency domain, by continuously introducing new data, the TARMA methods only need to modify the model parameters estimated in the previous step and can quickly model the system at the current moment [32]. This section is divided into the TARMAX model considering the actual milling force excitation, TARMAX model parameter recursive estimation, and sliding window recursive estimation.

### 2.1. TARMAX Model Considering Actual Milling Force Excitation

From the probability statistical chart, it can be found that the dynamic milling force signals were a superposition of the harmonic components and Gaussian white noise [18]. Therefore, the excitation exerted on the workpiece could be considered as a superposition of the theoretical milling force and Gaussian white noise. Subsequently, the time-varying dynamic equation of the machining system was expressed by an *n*-dimensional equation:(1)$$M(t)\ddot{x}(t)+C(t)\dot{x}(t)+K(t)x(t)=u(t)$$
where $M(t)\in R^{n\times n}$, $C(t)\in R^{n\times n}$, and $K(t)\in R^{n\times n}$ are the mass, damping, and stiffness matrices of the machining system, respectively. *t* is continuous time. $\ddot{x}(t)\in R^{n*1}$, $\dot{x}(t)\in R^{n*1}$, and $x(t)\in R^{n*1}$ are acceleration, velocity, and displacement vectors, respectively. $u(t)\in R^{n*1}$ is the input excitation signals, which contain theoretical milling force $F(t)\in R^{n*1}$ and the environmental excitation $e(t)\in R^{n*1}$, namely, ***u***(t) is defined as
(2)$$u(t)=F(t)+e(t)\;e(t)∼NID(0,\sigma^{2}(t))$$
where ***e***(t) is the environmental excitation, which can be treated as an uncorrelated white-noise excitation signal when the expectation and variance of the environmental excitation are 0 and **σ**^2^(*t*), respectively.

By continuously “freezing” the mass, damping, and stiffness matrices [33], the time-varying dynamic equation of Equation (1) can be transformed into TARMAX [32], which is given by
(3)$$x[t]+\sum_{i=1}^{N_{a}}a_{i}[t]x[t-i]=\sum_{j=0}^{N_{b}}b_{j}[t]u[t-j]=\sum_{j=0}^{N_{b}}b_{j}[t](F[t-j]+e[t-j])$$
where ***x***[*t*] is the discrete displacement response; ***u***[*t*] is the discrete excitation signal; ***F***[*t*] is the discrete milling force; ***e***[*t*] is the white noise sequence; ***a****_i_*[*t*] and ***b****_i_*[*t*] are the time-varying autoregressive coefficient and external excitation matrix, respectively; and *N*_a_ and *N*_b_ are the order of ***a****_i_*[*t*] and ***b****_i_*[*t*], respectively. Then, introducing backward shift operator *z*, and
(4)$$z^{i}x[t]=x[t-i]z^{i}$$

Substituting Equation (4) into Equation (3), Equation (3) can be rewritten in the state-space form
(5)A[z,t]x[t]=B[z,t]u[t]
where
(6)$$\left\{\begin{array}{l} A[z,t]=I+\sum_{i=1}^{N_{a}}a_{i}[t]z^{i}\\ B[z,t]=\sum_{j=0}^{N_{b}}b_{j}[t]z^{j} \end{array}\right.$$

Subsequently, the time-varying transfer function is obtained from Equation (5) as
(7)$$H[z,t]=A{\left[z,t\right]}^{-1}\circ B[z,t]$$

Defining $z=e^{-j\omega \Delta t}$, the “frozen time” transfer function of TARMAX is expressed as follows [34]
(8)$$H[\omega ,t]=A{\left[e^{-j\omega \Delta t},t\right]}^{-1}\circ B[e^{-j\omega \Delta t},t]\approx \frac{B[e^{-j\omega \Delta t},t]}{A[e^{-j\omega \Delta t},t]}$$
where j is an imaginary number and △*t* denotes the sampling period. Considering Equation (8), in every moment *t*, assume that $\left|A[e^{-j\omega \Delta t},t]\right|=0$, then, the “frozen time” transfer function poles ***p***_r_ (r = 1, …, *N*_a_) are determined, so the natural frequency of the continuous system Equation (1) can be calculated by [35]
(9)$$f_{r}[t]=\frac{\left|\ln p_{r}[t]\right|}{2\pi \cdot \Delta t}\;$$

For $\left|A[e^{-j\omega \Delta t},t]\right|=0$, i.e.,
(10)$$\left|A[e^{-j\omega \Delta t},t]\right|=\left|I+\sum_{i=1}^{N_{a}}a_{i}[t]z^{i}\right|=0$$

To avoid a non-linear system of Equation (10), the roots of Equation (10) can be transformed into a generalized eigenvalue problem [35], so Equation (10) can be rewritten as
(11)$$(D[t]-p_{r}[t]I)V_{r}[t]=0$$
where ***V***_r_[*t*] is an eigenvector, ***D***[*t*] is the eigenvalues matrix, which is defined as
(12)$$D[t]=\left[\begin{array}{cccc} 0 & I & \ldots & 0\\ \vdots & \vdots & \ddots & \vdots \\ 0 & 0 & \ldots & I\\ -a_{N_{a}} & -a_{N_{a}-1} & \ldots & -a_{1} \end{array}\right]$$

Therefore, if the time-varying autoregressive coefficients ***ɑ****_i_*[*t*] can be accurately estimated, and the natural frequency $f_{r}[t]$ of the time-varying dynamic system can be solved by combining Equations (9) and (11).

### 2.2. TARMAX Model Parameters Recursive Estimation

Based on the milling force model [36], the theoretical milling force can be predicted. However, according to Equation (3), it can be seen that ***ɑ****_i_*[*t*], ***b****_i_*[*t*], and ***e***[*t*] are unknown, and it is time-consuming to directly determine the ***ɑ****_i_*[*t*] matrix based on the TARMAX model. Therefore, in this section, the least square method is used to improve the calculation efficiency.

#### 2.2.1. Theoretical Milling Force Model

The natural frequencies in the *x* and *y* directions are lower than the natural frequencies in the *z* direction due to the clamping method and the machining mode. As the material is removed in milling, the workpiece is prone to chatter under the dynamic milling forces in the *x* and *y* directions. Then, in this paper, milling forces in the *x* and *y* directions are mainly considered and milling forces in the *z* direction are ignored. Therefore, in the milling of a thin-walled workpiece, the time-varying dynamic milling system can be expressed by an *n*-dimensional linear time periodic system, which is shown in Figure 1.

According to Figure 1, the dynamic milling force ***F***[*t*] is represented as follows [36]
(13)$$F\left[t\right]=aK_{c}\left(x\left[t\right]-x\left[t-T\right]\right)+af_{0}$$
where *a* is the depth of cut, *T* denotes the time delay period, ***f***_0_ expresses the static force and *K*_c_ is the milling force coefficients. In addition, ***f***_0_ and ***K****_c_* are defined as
(14)$$\left\{\begin{array}{l} K_{c}=\sum_{j=1}^{n_{1}}g\left(\phi_{j}\left[t\right]\right)\left[\begin{array}{cc} k_{xx} & k_{xy}\\ k_{yx} & k_{yy} \end{array}\right]\\ k_{xx}=-k_{tc}\sin\left(\phi_{j}\left[t\right]\right)\cos\left(\phi_{j}\left[t\right]\right)-k_{rc}\sin{\left(\phi_{j}\left[t\right]\right)}^{2}\\ k_{xy}=-k_{tc}\cos{\left(\phi_{j}\left[t\right]\right)}^{2}-k_{rc}\sin\left(\phi_{j}\left[t\right]\right)\cos\left(\phi_{j}\left[t\right]\right)\\ k_{yx}=k_{tc}\sin{\left(\phi_{j}\left[t\right]\right)}^{2}-k_{rc}\sin\left(\phi_{j}\left[t\right]\right)\cos\left(\phi_{j}\left[t\right]\right)\\ k_{yy}=k_{tc}\sin\left(\phi_{j}\left[t\right]\right)\cos\left(\phi_{j}\left[t\right]\right)-k_{rc}\cos{\left(\phi_{j}\left[t\right]\right)}^{2} \end{array}\right.$$
(15)$$f_{0}=\sum_{j=1}^{n_{1}}g\left(\phi_{j}\left[t\right]\right)\left[\begin{array}{l} f_{t}\left(\begin{array}{l} -k_{tc}\sin\left(\phi_{j}\left[t\right]\right)\cos\left(\phi_{j}\left[t\right]\right)-k_{rc}\sin{\left(\phi_{j}\left[t\right]\right)}^{2}\\ k_{tc}\sin{\left(\phi_{j}\left[t\right]\right)}^{2}-k_{rc}\sin\left(\phi_{j}\left[t\right]\right)\cos\left(\phi_{j}\left[t\right]\right) \end{array}\right)\\ +\left(\begin{array}{l} -k_{te}\cos\left(\phi_{j}\left[t\right]\right)-k_{re}\sin(\phi_{j}(t))\\ k_{te}\sin\left(\phi_{j}\left[t\right]\right)-k_{re}\cos\left(\phi_{j}\left[t\right]\right) \end{array}\right)\left(\phi_{j}\left[t\right]\right) \end{array}\right]$$
where *n*_1_ is number of teeth; *f*_t_ is feed per tooth in machining; *k_tc_*, *k**_r_**_c_*, *k_t_**_e_*, and *k**_re_* are the milling force coefficients; *g*(*ϕ*_j_[*t*]) is the cut-in function; and *ϕ*_j_[*t*] is the angular position of the j-th tooth; then, *ϕ*_j_[*t*] and *g*(*ϕ*_j_[*t*]) are
(16)$$\phi_{j}\left[t\right]=\left(2\pi \Omega /60\right)t+2\pi (j-1)/N$$
(17)$$g\left(\phi_{j}\left[t\right]\right)=\left\{\begin{array}{l} 1\;\mathrm{if}\;\phi_{st}<\phi_{j}\left[t\right]<\phi_{ex}\\ 0\;\mathrm{otherwise} \end{array}\right.$$
where Ω is the spindle speed, and *ϕ_st_* and *ϕ**_ex_* are the entry and exit angles on the *j*-th cutter tooth. For down milling, *ϕ_st_* = arccos(2*H*-1) and *ϕ**_ex_* = π, and for up milling, *ϕ_st_* = 0 and *ϕ**_ex_* = arccos(1-2*H*). *H* is the radial immersion ratio.

#### 2.2.2. Estimating Environmental Excitation ***ê***_1_[t]

Substituting Equation (2) into Equation (5) yields the following equation
(18)$$\begin{array}{l} A[z,t]x[t]=B[z,t](F[t]+e[t])\\ \Leftrightarrow G[z,t]x[t]=F[t]+e[t] \end{array}$$
where
(19)$$G[z,t]=B{\left[z,t\right]}^{-1}\circ A[z,t]$$

Theoretically, the excitation can be represented by an infinite order inverse function model [30], namely, Equation (18) can be rewritten as
(20)$$\sum_{i=0}^{\infty }g_{i}[t]x[t-i]=F[t]+e[t]$$

To estimate ***e***[*t*], the *N_g_* items of the inverse function model are taken as
(21)$$\begin{array}{l} \sum_{i=0}^{N_{g}}g_{i}[t]x[t-i]=F[t]+e[t]\\ \Leftrightarrow \left[g_{0}[t]g_{1}[t]\;\cdots g_{N_{g}}[t]\right]\left[\begin{array}{c} x[t]\\ x[t-1]\\ \vdots \\ x[t-N_{g}] \end{array}\right]=F[t]+e[t] \end{array}$$
with
(22)$$\left\{\begin{array}{l} G{\left[t\right]}^{T}=\left[g_{0}[t]g_{1}[t]\;\cdots g_{N_{g}}[t]\right]\\ \psi {\left[t\right]}^{T}=\left[x{\left[t-1\right]}^{T}x{\left[t-2\right]}^{T}\cdots x{\left[t-N_{g}\right]}^{T}\right] \end{array}\right.$$

Then, Substituting Equation (22) into Equation (21), Equation (21) can be reduced to
(23)$$G^{T}[t]\psi [t]=F[t]+e[t]$$
where ***G***^T^[*t*] and ***e***[*t*] are estimated using the least square method, and the estimation function of the least square method is given by
(24)$$\min\left[\frac{1}{2}\sum_{\tau =1}^{t}{\left\|G^{T}[t]\psi [\tau ]-F[\tau ]\right\|}^{2}\right]$$
where $\left\|•\right\|$ is the Euclidean norm. For this optimal problem, the solution of Equation (24) is calculated by
(25)$${\hat{G}}_{1}[t]={\left({\hat{\psi }}_{1}[t]{\hat{\psi }}_{1}{\left[t\right]}^{T}\right)}^{-1}{\hat{\psi }}_{1}[t]{\hat{F}}_{1}[t]$$
where
(26)$$\left\{\begin{array}{l} {\hat{\psi }}_{1}[t]=\left[\psi [1]\;\psi [2]\cdots \psi [t]\right]\;\\ {\hat{F}}_{1}[t]={\left[F[1]\;F[2]\cdots F[t]\right]}^{T} \end{array}\right.$$

Subsequently, substituting Equation (25) into Equation (23), the environmental excitation ***ê***_1_[*t*] can be estimated by
(27)$${\hat{e}}_{1}[t]={\hat{G}}_{1}{\left[t\right]}^{T}\psi [t]-F[t]$$

With the increase in iterative steps, the computational complexity of ${\left({\hat{\psi }}_{1}[t]{\hat{\psi }}_{1}{\left[t\right]}^{T}\right)}^{-1}$ progressively enlarges. To improve the computational efficiency of Equation (25), define the following
(28)$$P_{1}[t]={\left({\hat{\psi }}_{1}[t]{\hat{\psi }}_{1}{\left[t\right]}^{T}\right)}^{-1}$$

Next, the following equation can be received
(29)$$\begin{array}{l} P_{1}{\left[t\right]}^{-1}={\hat{\psi }}_{1}[t]{\hat{\psi }}_{1}{\left[t\right]}^{T}={\hat{\psi }}_{1}[t-1]{\hat{\psi }}_{1}{\left[t-1\right]}^{T}+\psi [t]\psi {\left[t\right]}^{T}\\ =P_{1}{\left[t-1\right]}^{-1}+\psi [t]\psi {\left[t\right]}^{T} \end{array}$$

Based on the matrix inverse theorem [34], ***P***_1_[*t*] can be rewritten as
(30)$$P_{1}[t]=\left(P_{1}[t-1]-\frac{P_{1}[t-1]\psi [t]\psi {\left[t\right]}^{T}P_{1}[t-1]}{I+\psi {\left[t\right]}^{T}P_{1}[t-1]\psi [t]}\right)$$

Substituting Equation (30) into Equation (25), we can obtain
(31)$$\begin{array}{l} {\hat{G}}_{1}[t]=\left(P_{1}[t-1]-\frac{P_{1}[t-1]\psi [t]\psi {\left[t\right]}^{T}P_{1}[t-1]}{I+\psi {\left[t\right]}^{T}P_{1}[t-1]\psi [t]}\right)\left({\hat{\psi }}_{1}[t-1]{\hat{F}}_{1}[t-1]+\psi [t]F{\left[t\right]}^{T}\right)\\ ={\hat{G}}_{1}[t-1]+\frac{P_{1}[t-1]\psi [t]\left(F{\left[t\right]}^{T}-\psi {\left[t\right]}^{T}{\hat{G}}_{1}[t-1]\right)}{I+\psi {\left[t\right]}^{T}P_{1}[t-1]\psi [t]}\\ ={\hat{G}}_{1}[t-1]+K_{1}[t] \end{array}$$
where
(32)$$K_{1}[t]=\frac{P_{1}[t-1]\psi [t]\left(F{\left[t\right]}^{T}-\psi {\left[t\right]}^{T}{\hat{G}}_{1}[t-1]\right)}{I+\psi {\left[t\right]}^{T}P_{1}[t-1]\psi [t]}$$

From Equation (31), it is found that the relationship between ***Ĝ***_1_[*t*-1] and ***Ĝ***_1_[*t*] can be established. Therefore, the calculation volume of Equations (31) and (27) is reduced by iterative calculations.

#### 2.2.3. Estimating Coefficient Matrices ***ŵ***_1_[t]

Substituting Equation (27) into Equation (3), the time-varying TARMAX dynamic equation can be expressed as
(33)$$\begin{array}{l} x[t]+\sum_{i=1}^{N_{a}}a_{i}[t]x[t-i]=\sum_{j=0}^{N_{b}}b_{j}[t]({\hat{e}}_{1}[t-j]+F[t-j])\\ \Leftrightarrow x[t]=w^{T}[t]\varphi [t] \end{array}$$
where
(34)$$\left\{\begin{array}{l} w[t]={\left[a_{1}[t]\cdots a_{N_{a}}[t]b_{0}[t]\cdots b_{N_{b}}[t]\right]}^{T}\\ \varphi {\left[t\right]}^{T}=\left[-x{\left[t-1\right]}^{T}\cdots -x{\left[t-N_{a}\right]}^{T}F{\left[t\right]}^{T}+{\hat{e}}_{1}{\left[t\right]}^{T}\cdots F{\left[t-N_{b}\right]}^{T}+{\hat{e}}_{1}{\left[t-N_{b}\right]}^{T}\right] \end{array}\right.$$

Then, the coefficient matrices ***ŵ***_1_[*t*] can be estimated using the least square method, and
(35)$${\hat{w}}_{1}[t]={\left({\hat{\varphi }}_{1}[t]{\hat{\varphi }}_{1}{\left[t\right]}^{T}\right)}^{-1}{\hat{\varphi }}_{1}[t]{\hat{X}}_{1}[t]$$
where
(36)$$\left\{\begin{array}{l} {\hat{\varphi }}_{1}[t]=\left[\varphi [1]\;\varphi [2]\cdots \varphi [t]\right]\\ {\hat{X}}_{1}[t]={\left[x[1]\;x[2]\cdots x[t]\right]}^{T} \end{array}\right.$$

Subsequently, combining Equations (34) and (35), the time-varying coefficient matrices ***a****_i_*[*t*] and ***b****_i_*[*t*] in the TARMAX model are obtained. Using the same simplified method in Equation (25) for Equation (35), defines the following
(37)$$P_{2}[t]={\left({\hat{\varphi }}_{1}[t]{\hat{\varphi }}_{1}{\left[t\right]}^{T}\right)}^{-1}$$

Simplified in the same way as Equation (31), Equation (35) can be reduced to
(38)$${\hat{w}}_{1}[t]={\hat{w}}_{1}[t-1]+K_{2}[t]$$
where
(39)$$\begin{array}{l} P_{2}[t]=\left(P_{2}[t-1]-\frac{P_{2}[t-1]\varphi [t]\varphi {\left[t\right]}^{T}P_{2}[t-1]}{I+\varphi {\left[t\right]}^{T}P_{2}[t-1]\varphi [t]}\right)\\ K_{2}[t]=\frac{P_{2}[t-1]\varphi [t]\left(x{\left[t\right]}^{T}-\varphi {\left[t\right]}^{T}{\hat{w}}_{1}[t-1]]\right)}{I+\varphi {\left[t\right]}^{T}P_{2}[t-1]\varphi [t]} \end{array}$$

### 2.3. TARMAX Model Parameters Sliding Windows Recursive Estimation

According to Equations (26) and (36), it can be found that the dimensions of ${\hat{\psi }}_{1}[t]$ and ${\hat{\varphi }}_{1}[t]$ gradually increase with time, and the amount of calculations for ***e***[*t*] and ***a****_i_*[*t*] will multiply. Therefore, to simplify the calculation process, a sliding window *N* is introduced. When *t* > *N*, the estimation of ***e***[*t*] and ***a****_i_*[*t*] only depends on the latest data. Then, ***e***[*t*] and ***a****_i_*[*t*] are calculated using the following method.

#### 2.3.1. Estimating Environmental Excitation ***ê***_2_[t]

To improve the calculation efficiency, the latest data *N* are applied for the least square estimation function. Therefore, the estimation function of the sliding window least square method is used for ***ê***_2_[*t*] and is defined as
(40)$$\min\left[\frac{1}{2}\sum_{\tau =t-N+1}^{t}{\left\|G^{T}[t]\psi [\tau ]-F[\tau ]\right\|}^{2}\right]$$

For Equation (40), the solution is given by
(41)$${\hat{G}}_{2}[t]={\left({\hat{\psi }}_{2}[t]{\hat{\psi }}_{2}{\left[t\right]}^{T}\right)}^{-1}{\hat{\psi }}_{2}[t]{\hat{F}}_{2}[t]$$
where
(42)$$\left\{\begin{array}{l} {\hat{\psi }}_{2}[t]=\left[\psi [t-N+1]\;\psi [t-N+2]\cdots \psi [t]\right]\\ {\hat{F}}_{2}[t]={\left[F[t-N+1]\;F[t-N+2]\cdots F[t]\right]}^{T} \end{array}\right.$$

Subsequently, considering Equations (23) and (41), the environmental excitation ***ê***_2_[*t*] can be expressed by
(43)$${\hat{e}}_{2}[t]={\hat{G}}_{2}{\left[t\right]}^{T}\psi [t]-F[t]$$

To reduce the computational effort of Equation (41), use the following
(44)$$\left\{\begin{array}{l} P_{3}{\left[t\right]}^{-1}={\hat{\psi }}_{2}[t]{\hat{\psi }}_{2}{\left[t\right]}^{T}=P_{3}{\left[t-1\right]}^{-1}+U_{1}[t]U_{1}{\left[t\right]}^{T}\\ U_{1}[t]=[\psi [t]j\psi [t-N]] \end{array}\right.$$

Similarly, considering matrix inverse theorem [34], ***P***_3_[*t*] can be converted to
(45)$$P_{3}[t]=\left(P_{3}[t-1]-\frac{P_{3}[t-1]U_{1}[t]U_{1}{\left[t\right]}^{T}P_{3}[t-1]}{I+U_{1}{\left[t\right]}^{T}P_{3}[t-1]U_{1}[t]}\right)$$

Then, substituting Equation (45) into Equation (41) yields
(46)$$\begin{array}{l} {\hat{G}}_{2}[t]=\left(P_{3}[t-1]-\frac{P_{3}[t-1]U_{1}[t]U_{1}{\left[t\right]}^{T}P_{3}[t-1]}{I+U_{1}{\left[t\right]}^{T}P_{3}[t-1]U_{1}[t]}\right)\left({\hat{\psi }}_{2}[t-1]{\hat{F}}_{2}[t-1]+U_{1}[t]\bar{F}{\left[t\right]}^{T}\right)\\ ={\hat{G}}_{2}[t-1]+K_{3}[t] \end{array}$$
where
(47)$$\left\{\begin{array}{l} K_{3}[t]=\frac{P_{3}[t-1]U_{1}[t]\left(\bar{F}{\left[t\right]}^{T}-U_{1}{\left[t\right]}^{T}{\hat{G}}_{2}[t-1]\right)}{I+U_{1}{\left[t\right]}^{T}P_{3}[t-1]U_{1}[t]}\\ \bar{F}[t]=\left[F[t]\;jF[t-N]\right] \end{array}\right.$$

#### 2.3.2. Estimating Coefficient Matrices ***ŵ***_2_[t]

Based on Equations (3) and (43), the estimation method in this section is similar to that of ***ŵ***_1_[*t*]. Therefore, the parameter ***ŵ***_2_[*t*] can be estimated using the least square method, and
(48)$${\hat{w}}_{2}[t]={\left({\hat{\varphi }}_{2}[t]{\hat{\varphi }}_{2}{\left[t\right]}^{T}\right)}^{-1}{\hat{\varphi }}_{2}[t]{\hat{X}}_{2}[t]$$
where
(49)$$\left\{\begin{array}{l} {\hat{\varphi }}_{2}[t]=\left[\varphi [t-N+1]\;\varphi [t-N+2]\cdots \varphi [t]\right]\\ {\hat{X}}_{2}[t]={\left[x[t-N+1]\;x[t-N+2]\cdots x[t]\right]}^{T} \end{array}\right.$$

Defined
(50)$$\left\{\begin{array}{l} P_{4}{\left[t\right]}^{-1}={\hat{\varphi }}_{2}[t]{\hat{\varphi }}_{2}{\left[t\right]}^{T}=P_{4}{\left[t-1\right]}^{-1}+U_{2}[t]U_{2}{\left[t\right]}^{T}\\ U_{2}[t]=\left[\varphi [t]\;j\varphi [t-N]\right] \end{array}\right.$$

Similarly, Equation (48) can be reduced to
(51)$${\hat{w}}_{2}[t]={\hat{w}}_{2}[t-1]+K_{4}[t]$$
where
(52)$$\left\{\begin{array}{l} P_{4}[t]=\left(P_{4}[t-1]-\frac{P_{4}[t-1]U_{2}[t]U_{2}{\left[t\right]}^{T}P_{4}[t-1]}{I+U_{2}{\left[t\right]}^{T}P_{4}[t-1]U_{2}[t]}\right)\\ K_{4}[t]=\frac{P_{4}[t-1]U_{2}[t]\left(\bar{X}{\left[t\right]}^{T}-U_{2}{\left[t\right]}^{T}{\hat{w}}_{2}[t-1]]\right)}{I+U_{2}{\left[t\right]}^{T}P_{4}[t-1]U_{2}[t]}\\ \bar{X}[t]=\left[x[t]\;jx[t-N]\right] \end{array}\right.$$

In addition, during calculation, false modes may appear, which may lead to the natural frequency estimation deviation. Therefore, to avoid false modes, a judging function *J*(·) is defined as
(53)$$J({\hat{f}}_{r}[t])=\left\{\begin{array}{l} {\hat{f}}_{r}[t-1]\;\mathrm{if}\;\left|\left({\hat{f}}_{r}[t]-{\hat{f}}_{r}[t-1]\right)/{\hat{f}}_{r}[t]\right|>\zeta \\ {\hat{f}}_{r}[t]\;\mathrm{otherwise} \end{array}\right.$$
where ζ is the threshold value. If the relative error between the natural frequency at moment *t* and that at the moment *t*-1 is greater than the threshold value, then, the output natural frequency at moment *t* will be replaced by that at moment *t*-1.

From above calculation, when *t* < *N*, the environment excitation ***ê***_1_[*t*] can be determined by Equations (27) and (31), and the time-varying coefficient ***ŵ***_1_[*t*] can be obtained by Equation (38). Nevertheless, when *t* > *N*, the environment excitation ***ê***_2_[*t*] be determined by Equations (43) and (46), and the time-varying coefficient ***ŵ***_2_[*t*] can be obtained by Equation (51). In addition, the time-varying autoregressive coefficient ***a***_i_[*t*] can be extracted from ***ŵ***_1_[*t*] or ***ŵ***_2_[*t*] by Equation (34). Finally, the system modal parameters can be calculated by Equation (9), and Equations (11) and (53). Then, the flow chart of the proposed algorithm in milling is shown in Figure 2.

## 3. Numerical Simulation Verification

In this section, several numerical simulation experiments were used to verify the effectiveness of the proposed method. As the natural frequency of the first few orders of a thin-walled part has a large influence on the milling stability, the same three-degree-of-freedom structure as shown in the literature [18,37,38] was used to calculate the response of a thin-walled part subjected to a milling force excitation, which is shown in Figure 3. Then, the parameters were selected for numerical simulation and are shown in Table 1. To analyze the system dynamic characteristics, three theoretical cutting forces were calculated, and 30 sets of white-noise signals were generated for simulation using the Monte Carlo method according to the parameters in Table 1. Subsequently, based on the “frozen time” method, the simulated time-varying frequency response functions and the natural frequencies of the 3-DOF time-varying structure were obtained, which are shown in Figure 4 and Figure 5, respectively.

From Figure 3 and Table 1, it can be seen that the system is excited by the independent milling force signals (theoretical milling force signal and Gaussian white noise signal). From Figure 4, the frequency response functions are obtained under excited conditions, but the natural frequency values of the system are approximately equal. In addition, from Figure 5, the first, the second, and the third natural frequencies are extracted, and we can see that the natural frequencies gradually reduced with time.

Subsequently, for the numerical simulation system, the response under a theoretical cutting force and Gaussian white noise was obtained in order for identifying the system modal parameters. In this process, 30 sets of displacement responses were obtained, then, to closely simulate the actual cutting condition, white noise signals with signal-to-noise ratios (SNR) of 15 dB, 20 dB, and 30 dB were added to contaminate the obtained displacement response signals; a set of displacement response signals with SNR = 15 dB is shown in Figure 6. Then, the cutting displacement responses contaminated with 15 dB, 20 dB, and 30 dB were used to calculate and determine the modal parameters of the simulated system.

To reduce the effect of initialization errors on the calculation results, the initial 500 samples (0.2 s) were discarded. Subsequently, the TARMAX model for the sliding window lengths of *N* = 50, *N* = 100, and *N* = 200 was used to estimate the system modal parameters, which were compared with the benchmark theoretical frequency and are shown in Figure 7.

From Figure 7, it was found that the estimated natural frequency presented good tracking results. As the SNR decreased, many scattered points deviated from the theoretical value. Under the constraint of Equation (53), the estimated natural frequencies fluctuated within the allowable error range, which validated the effectiveness of the proposed method. Then, when the sliding window length was relatively small, the natural frequencies were well estimated compared with the theoretical value, but fluctuated obviously. When the sliding window length was large, all of the estimated natural frequencies agreed well with the theoretical value, and fewer false modals appeared. In addition, it could be seen that the second-order natural frequency was not adequately tracked at SNR = 15 due to the large sliding window length, and the existence of some errors.

To further validate the proposed method, the mean absolute error (MAE) was introduced and is defined as
(54)$$\mathrm{MAE}=\frac{1}{R}\sum_{i=1}^{R}\frac{1}{L}\sum_{t=1}^{L}\left|f[t]-{\hat{f}}_{i}[t]\right|.$$
where ***f***[*t*] is the theoretical natural frequency, ${\hat{f}}_{i}[t]$ denotes the estimated natural frequency in the *i*-th Monte Carlo experiment, and *L* is the data length. Based on this, the different mean absolute error (MAE) was calculated and is shown in Table 2. From Table 2, with the increase in sliding window length, the prediction accuracy of the proposed method was obviously improved. Especially, when *N* = 200, the estimated accuracy was higher at the different SNR, and the MAE of the proposed method was also significantly smaller than that of the other sliding window lengths, which could indicate that the noise pollution was robust. Therefore, the proposed method could be widely used to identify the modal parameters under the data noise pollution, and be applied to practical conditions.

## 4. Experimental Validation and Discussion

In order to verify the effectiveness of the proposed method in the milling of thin-walled plates, several experiments were conducted. In the experiments, the materials of the plate and cutter were TC4 and cemented carbide, respectively, and the material parameters are shown in Table 3. The size of the plate used in the experiments was 80 × 40 × 3 mm. In addition, all of the experiments were carried out on a three-axis machine center (VMC-850E), the dynamic cutting forces in the milling were measured using a Kistler9257B, and the modal parameters were measured using a model hammer (500 N), an acquisition instrument DH5981, and acceleration sensors (ref. sensitivity 10.25 mV/g). The machining and modal test setups are shown in Figure 8.

### 4.1. Milling Force Coefficients Identification

To validate the effectiveness of the proposed method, the predicted cutting forces were needed. Therefore, the cutting force coefficients needed to be calibrated. Then, five slotting tests were carried out with a spindle speed of 1000 r/min and an axial depth of cut of 0.5 mm, while the feed rates were 40 mm/min, 80 mm/min, 120 mm/min, 160 mm/min, and 200 mm/min. Then, the milling force coefficient and the average milling force were obtained [36,39].
(55)$$\left\{\begin{array}{l} {\bar{F}}_{x}=-\frac{na}{4}k_{rc}f_{t}-\frac{na}{\pi }k_{re}\\ {\bar{F}}_{y}=\frac{na}{4}k_{tc}f_{t}+\frac{na}{\pi }k_{te}\\ {\bar{F}}_{z}=\frac{na}{\pi }k_{ac}f_{t}+\frac{na}{2}k_{ae} \end{array}\right.$$

Forces in the *x* and *y* directions were mainly considered in the model. In addition, the actually tested milling forces in the *z*-direction were small and easily contaminated by noise. Therefore, the cutting force coefficient in the *z* direction was not identified. Subsequently, the cutting force coefficients in the *x* and *y* directions were identified as follows *k*_tc_ = 1120.8 N/mm^2^, *k*_rc_ = 2285.6 N/mm^2^, *k*_te_ = 9.16 N/mm, and *k*_re_ = 13.21 N/mm.

### 4.2. Modal Parameters Evolution Considering Material Removal

To analyze the evolution of the modal parameters caused by the material removal, different impact experiments were conducted after five milling stages. In the experiments, the impact tests were conducted to assess the workpiece modal parameters before and after each machining stage. The experimental setups contained the acquisition apparatus DH5981, the accelerometer (5000 g, sensitivity 10.25 mV/g, and mass 5 g), and the modal hammer (500 N, sensitivity 1 mV/N). Then, the dynamic responses were changing at different impact positions and machining stages. Therefore, taking into account the fixture constraints, three points (point 1, point 2, and point 3) on a thin plate were selected for the tests, which are shown in Figure 9a, where it can be seen that point 2 is located in the middle of the edge of the plate, and point 1 and point 3 are symmetrical with respect to point 2. Subsequently, the frequency response functions on different impact measured points can be obtained under different machining stages, and it was found that the frequency response functions were almost the same for point 1 and point 3. However, considering the position of point 1 and point 3 on the plate, an unstable vibration signal caused by low rigidity would be generated. Therefore, point 2 was chosen for the measured response. Then, the sensor position and the machining region were determined and are shown in Figure 9b,c.

Next, five groups of milling experiments were conducted and the material removal patterns are shown in Figure 10. The selected cutting parameters were a spindle speed of 2000 rpm, axial depth of cut of 5 mm, radial depth of cut of 0.2 mm, feed rate of 120 mm/min, down milling, and the vibration responses in different cutting stages were obtained and are shown in Figure 11. After every cut, each impact test after milling was carried out, and the frequency response functions were obtained, which are shown in Figure 12. Then, from Figure 11, it can be seen that the acceleration response of the workpiece was unstable in the cut-in and cut-out stages due to the structural properties and rigidity. When the tool was fully cut into a workpiece, the coupled effect of workpiece–tool was weak, so the vibration was stable. As the number of cutting layers increased, the acceleration response was progressively reduced. The possible reasons for this are as follows: In the first cut, chatter occurred, which caused a larger acceleration response. However, as the material was removed, the modal parameters were time-varying in the milling and used the same machining parameters for the next milling, the chatter gradually decreased, which led to a low acceleration response. Subsequently, from Figure 12, we can find that with the workpiece material removal, the modal parameters of the machining system were significantly reduced from 525 Hz, 501 Hz, 504 Hz, 482 Hz, 465 Hz, to 453 Hz. Therefore, it is necessary to investigate the time-varying modal parameters for improving the stability.

### 4.3. Model Validation and Discussion

To identify the modal parameters of a machining system, a displacement response should be used for the proposed method. Therefore, a program by MATLAB was applied to convert the vibration acceleration responses measured in the milling to the vibration displacement responses, and the process is as follows.

Without a loss of generality, the acceleration signal $\ddot{x}(t)$ in the frequency domain can be expressed as follows
(56)$$\ddot{x}(t)=Ae^{j\omega t}$$
where ***A*** is the coefficient corresponding to $\ddot{x}(t)$. Based on the inverse Fourier transform theory, assuming that the initial velocity and displacement of the system are 0, the displacement signal ***x***[*t*] can be obtained using Equation (56).
(57)$$x[t]=\sum_{k=0}^{N-1}-\frac{1}{{(2\pi k\Delta f)}^{2}}H[k]x[k{]e}^{2j\pi kr/N}$$
where $X[k]=-A/\omega_{k}^{2}$, $\omega_{k}=k\Delta f$, *H*[k] denotes the cut-off frequency, which is defined as
(58)$$H[k]=\left\{\begin{array}{l} 1\;(f_{d}\leq k\Delta f\leq f_{u})\\ 0\;(\mathrm{otherwise}) \end{array}\right.$$

Based on Equations (56)–(58), the displacement response of the machining system after five cuts was calculated and is shown in Figure 13.

From Figure 11 and Figure 13, the vibration responses obviously fluctuated in the cut-in and cut-out stages, with the reason for this being that the dynamic cutting force was higher and the rigidity of the plate was lower in the cut-in and cut-out stages, respectively. For example, at the beginning and end of the fourth group of experiments, there was a significant change in the displacement signal. Therefore, to avoid anomalous data interference, the relatively stable response data over a 16 s period (from 3.3 s to 19.3 s) was chosen to identify the system modal parameters. Considering the actual milling process, the first order natural frequencies were closely associated with the system machining stability, which was validated by many references. In addition, as the modal parameters during machining were not available in real time, it was assumed in the paper that the natural frequency of the workpiece varied linearly during each experiment. Finally, the first order natural frequencies were calculated by TARMAX (*N* = 200) in different states, and the predicted results are shown in Figure 14.

From Figure 14, it was found that the predicted natural frequencies using the proposed method agreed well with the experimental values. The experimental average of the five tests was 513, 502.5, 493, 473.5, and 459 Hz and the predicted average of the five tests was 516.16, 506.24, 496.45, 479.31, and 462.71 Hz. The predicted values were all higher than the experimental values due to the coupling benefits of the tool and the workpiece during machining, which is consistent with the experimental phenomena in the literature [15]. In addition, several scatter points offset the baseline under the machining start and end stage. The reason for this is that the milling was at the beginning or end of cut at this time, and the vibration displacement response obviously changed, which caused a large deviation between the predicted values and the experimental values.

To further demonstrate the validity of the proposed method, the mean absolute error (MAE) and mean relative error (MRE) were introduced, and are defined as
(59)$$\left\{\begin{array}{l} \mathrm{MAE}=\frac{1}{L}\sum_{t=1}^{L}\left|f[t]-\hat{f}[t]\right|\\ \mathrm{MRE}=\frac{1}{L}\sum_{t=1}^{L}\frac{\left|f[t]-\hat{f}[t]\right|}{f[t]} \end{array}\right.$$
where ***f***[*t*] is the theoretical natural frequency, $\hat{f}[t]$ denotes the estimated natural frequency, and *L* is the data length. Based on this, MAE and MRE were calculated and shown in Figure 15.

From Figure 15, the average absolute error of the TARMAX method was around 20 Hz and the relative error was around 4.3%, and the prediction results were relatively stable. The third group experiments had the highest prediction accuracy because smoother displacement signals were chosen. Next, there were some non-smooth signals in the displacement signals in the other sets of experiments. As the number of milling experiments increased, the relative error of the TARMAX method showed an upward trend. The possible reasons for this are as follows. As the number of experiments increased, the tool wore out and the dynamic milling forces changed, thus causing fluctuations in the vibration response, which is consistent with the observation that the displacement signals in the fourth and fifth experiment included some non-stationary components. In addition, the calculation time and complexity of the TARMAX method are mainly relevant to the sliding window length, which makes the proposed method handle vibration responses online better. Overall, the TARMAX method can commendably predict the modal parameters in the milling of a thin-walled workpiece.

## 5. Conclusions

In the milling of a thin-walled workpiece, the dynamic characteristics of the thin-walled workpiece-fixture system are time-varying due to material removal. To investigate the time-varying characteristics of the modal parameters in milling, an accurate TARMAX identification method under external excitation was proposed. The proposed method can rapidly and precisely identify the modal parameters in the milling of thin-walled workpieces. Thus, the contributions of the paper are as follows.

(1)The TARMAX modal parameter identification method under external excitation was proposed. In this process, the dynamic modal parameters are estimated through the measured vibration response in milling using the recursive estimation method or sliding windows recursive estimation.(2)A 3-DOF time-varying structure model under milling forces and white noise excitation was established. The predicted natural frequencies using the proposed method were in close agreement with the simulated values.(3)The proposed model is validated by the numerical simulation and machining experiments. Moreover, the prediction accuracy of the TARMAX identification method was 95.68%, and good agreement was drawn in the numerical simulation and machining experiments. Our further research objectives will focus on the prediction of damping ratios, modal stiffness, and modal mass.

## Figures and Tables

**Figure 1 micromachines-13-01581-f001:**
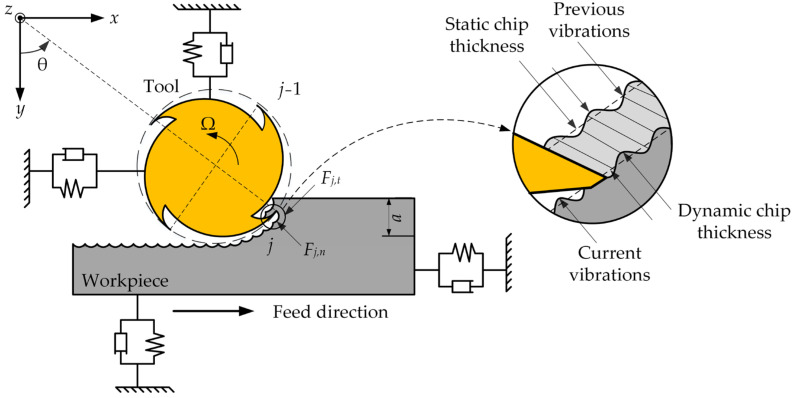
Dynamic milling dynamics model.

**Figure 2 micromachines-13-01581-f002:**
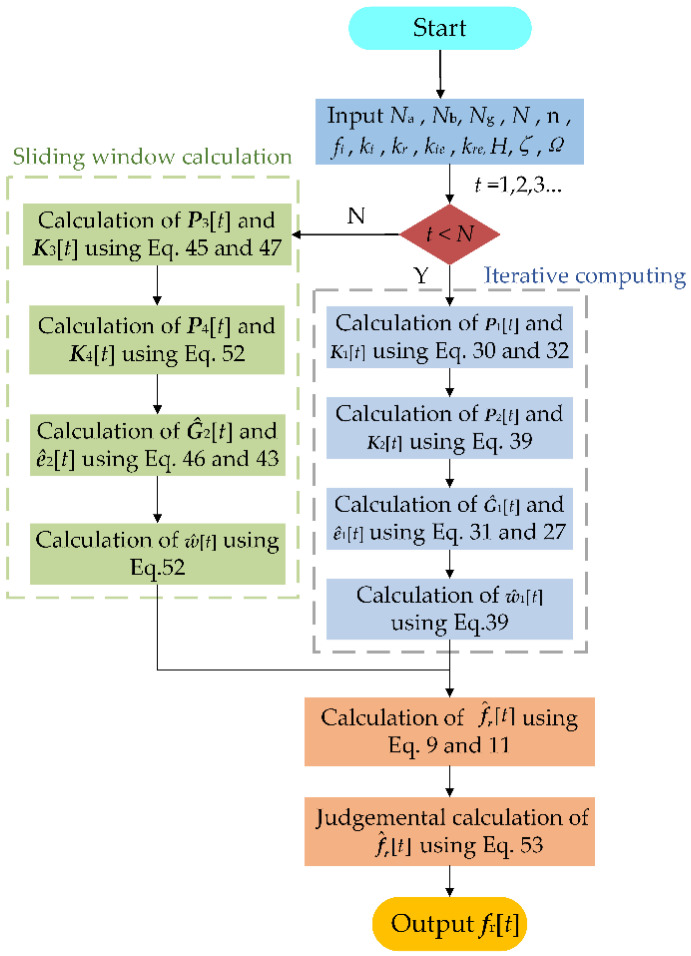
Flow chart of the algorithm for estimating the modal parameters.

**Figure 3 micromachines-13-01581-f003:**
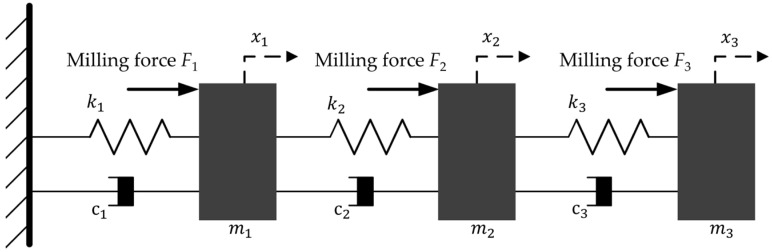
The 3-DOF time-varying structure excited by milling forces.

**Figure 4 micromachines-13-01581-f004:**
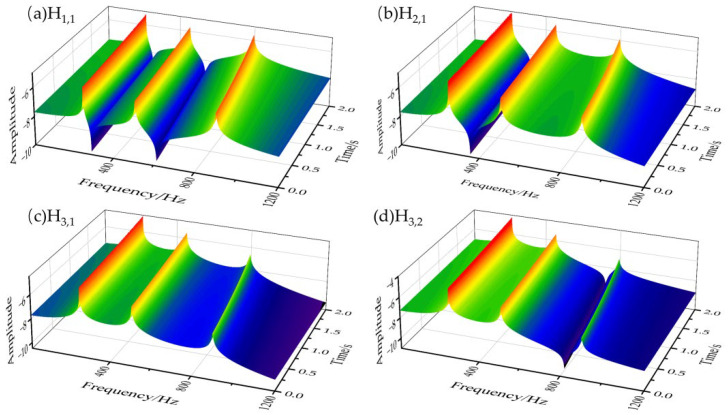
Frequency response functions of the 3-DOF time-varying structure.

**Figure 5 micromachines-13-01581-f005:**
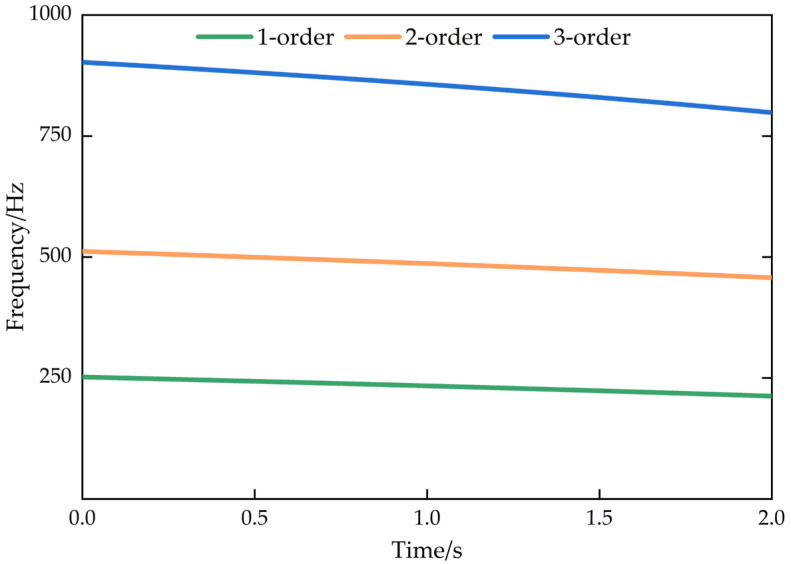
Natural frequencies of the 3-DOF time-varying structure.

**Figure 6 micromachines-13-01581-f006:**
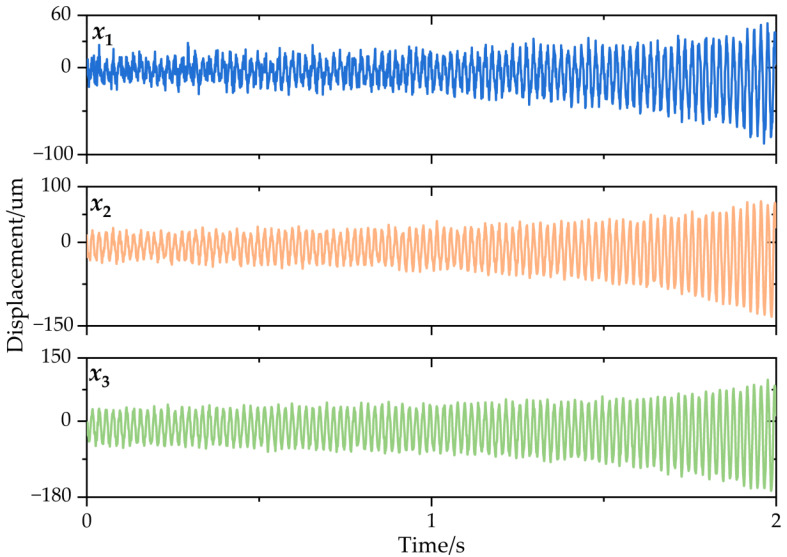
Displacement response of the 3-DOF time-varying structure.

**Figure 7 micromachines-13-01581-f007:**
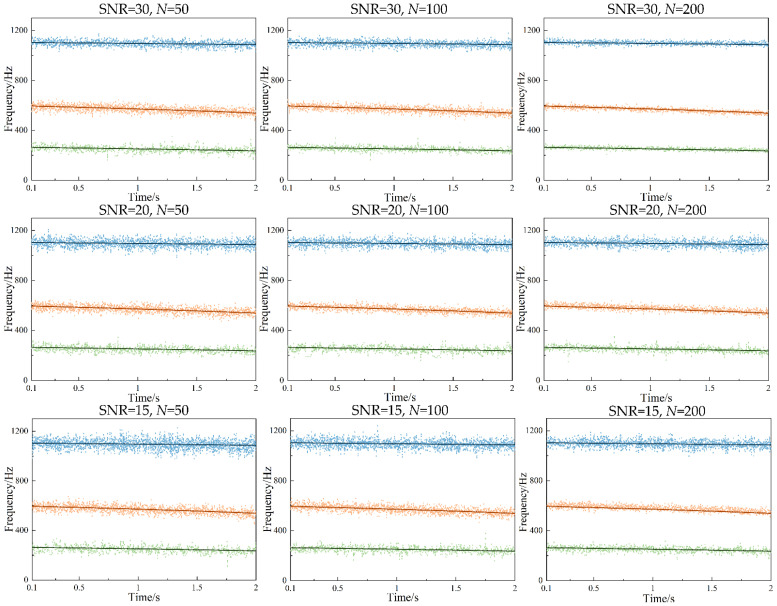
Natural frequencies by TARMAX under different SNR and different sliding window lengths; solid lines: estimated values; dotted lines: simulated values.

**Figure 8 micromachines-13-01581-f008:**
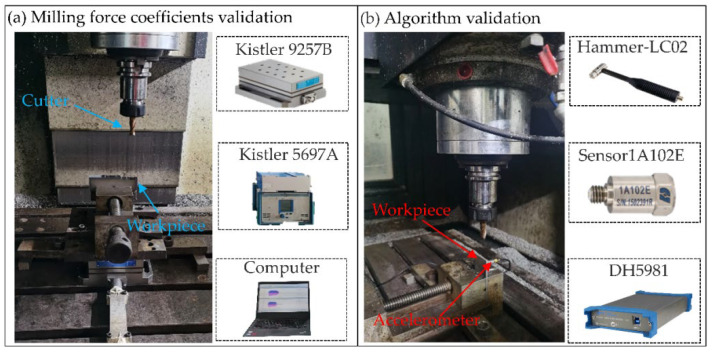
Machining and modal test.

**Figure 9 micromachines-13-01581-f009:**
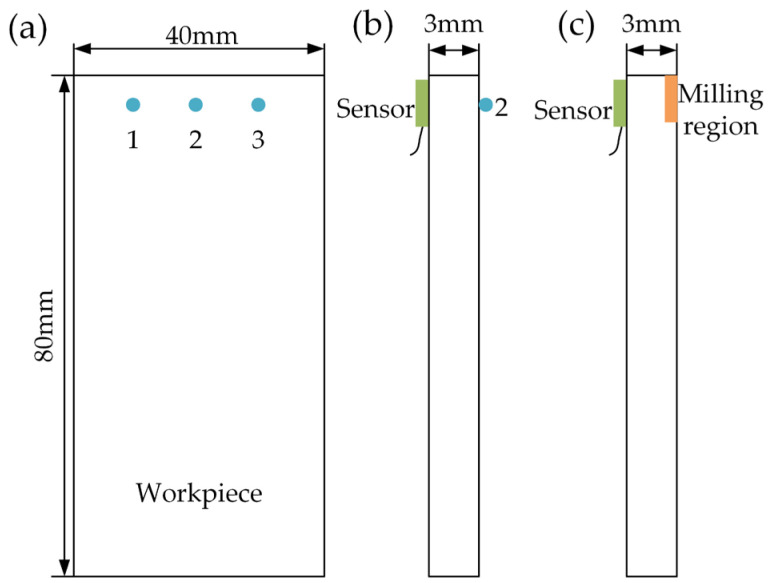
Experimental design. (**a**) Distributed points 1, 2, and 3 on the thin-walled plate for the impact experiments. (**b**) Position of the sensor and point 2 used to measure the response. (**c**) Position of the sensor and tool region for machining.

**Figure 10 micromachines-13-01581-f010:**
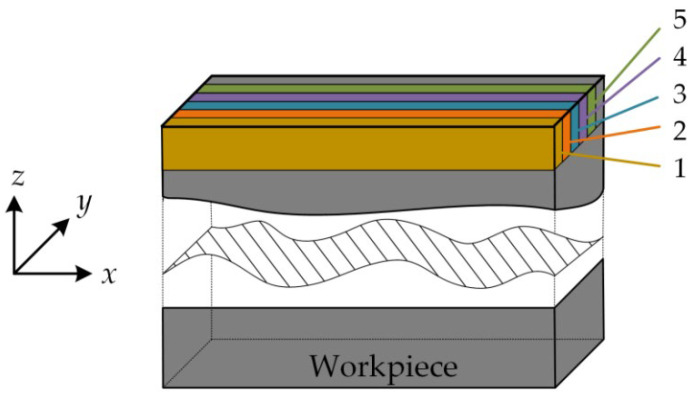
Material removal patterns in milling.

**Figure 11 micromachines-13-01581-f011:**
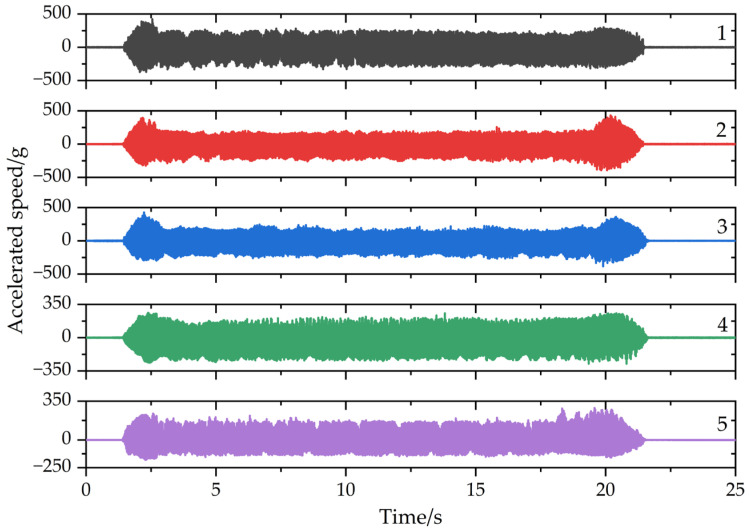
Acceleration responses in different milling stages.

**Figure 12 micromachines-13-01581-f012:**
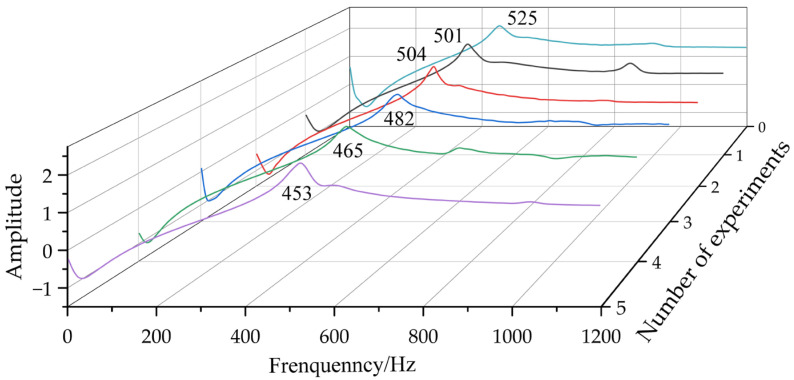
Frequency response functions in different milling stages.

**Figure 13 micromachines-13-01581-f013:**
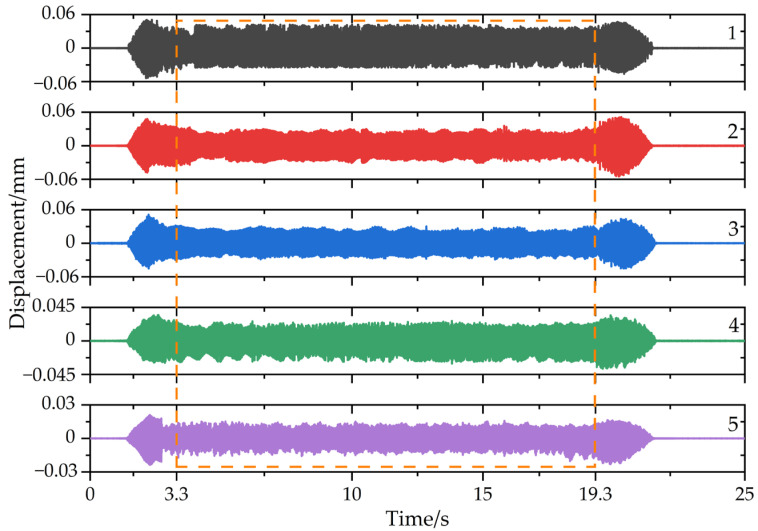
Workpiece displacement responses in different milling stages.

**Figure 14 micromachines-13-01581-f014:**
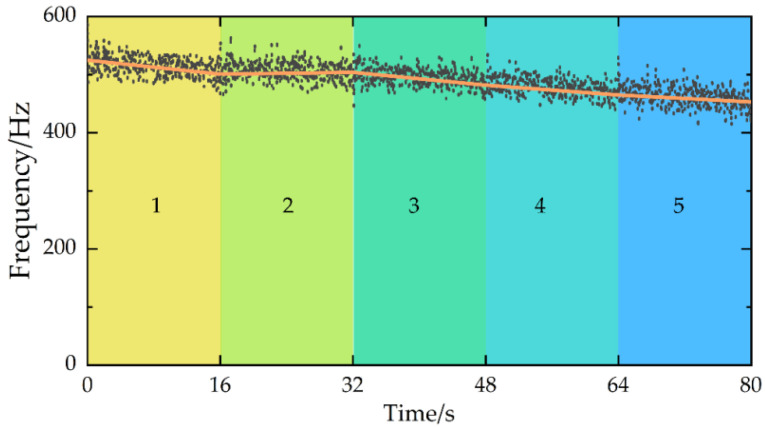
First-order natural frequency predicted results in different milling stages; line: experimental values; dot: estimated values.

**Figure 15 micromachines-13-01581-f015:**
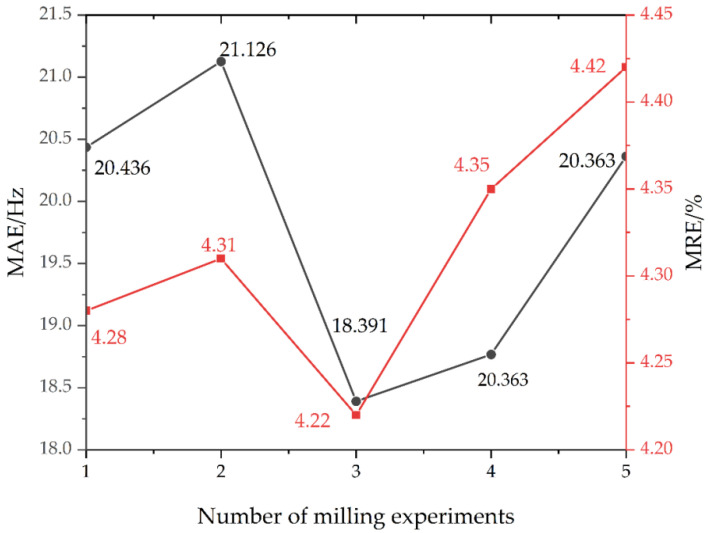
Error between the predicted results and true values; black line: MAE; red line: MRE.

**Table 1 micromachines-13-01581-t001:** Simulation parameters for the 3-DOF time-varying structure [39].

Parameters	Values
Mass, stiffness and damping	*m*_1_ = 3 − 0.3*t*, *m*_2_ = 3 − 0.1*t*, *m*_3_ = 2 *k*_1_ = 5 × 10^7^ − 10^7^*t*, *k*_2_ = 3 × 10^7^ − 5 × 10^6^*t*, *k*_3_ = 10^7^ − 10^6^*t*, *c*_1_ = 200, *c*_2_ = 100, *c*_3_ = 50,
Variances of white noise signal	$\sigma_{1}^{2}$ = 4, $\sigma_{2}^{2}$ = 2, $\sigma_{3}^{2}$= 1
Process parameters	Ω = 1500, m = 40, *ϕ*_ex_ = 0, *ϕ*_ts_ = π, *n*_1_ = 2, *a* = 2 × 10 − 4, *f*_t_ = 10 − 3
Milling force coefficients	*k*_tc1_ = 6 × 10^7^, *k*_rc1_ = 2 × 10^7^, *k*_te1_ = 3 × 10^3^, *k*_re1_ = 10^3^, *K*_tc__2_ = *k*_tc3_ = 8 × 10^7^, *k*_te2_ = *k*_te3_ = 6 × 10^3^, *K*_rc__2_ = *k*_rc3_ = 3 × 10^7^, *k*_re2_ = *k*_re3_ = 4 × 10^3^

**Table 2 micromachines-13-01581-t002:** Modal parameter identification errors under different external excitations.

Noise	Mean Absolute Error/Hz		
	*N* = 50	*N* = 100	*N* = 200
SNR = 30	18.9807	16.3046	10.2242
SNR = 20	22.2530	20.2492	16.0250
SNR = 15	26.1829	22.6237	19.0390

**Table 3 micromachines-13-01581-t003:** Workpiece and tool parameters.

Workpiece	Density	Poisson Ratio	Young’s Modulus	Materials
	4.6 g/cm3	0.34	108 GPa	TC4
Cutter	Diameter	Number of teeth	Spiral angle	Length
	12 mm	2	30°	75 mm

## Data Availability

The data that support the findings of this study are available from corresponding author upon reasonable request.

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
