# Peer review of "Output-Only Time-Varying Modal Parameter Identification Method Based on the TARMAX Model for the Milling of a Thin-Walled Workpiece"

_micromachines, 2022, doi:10.3390/mi13101581_

Round 1

Reviewer 1 Report

1. The method proposed in this paper is used to obtain the in-process modal parameters of thin-walled parts, However, there are only prediction data of natural frequency in the prediction result. Is it possible to supplement the predicted results for the magnitude of the frequency response function?

2. Please confirm that the description of the cutting force in Eq. 15 is correct. Please explain why only the cutting forces in the x and y directions are applied and the cutting forces in the z direction are ignored.

3. In the page 15, please confirm that the identified cutting force coefficients are correct. In general, the tangential shear force coefficient ktc should be larger than the radial shear force coefficient krc.

4. The error analysis between the predicted results and the experimental results is carried out in the article. Please list the predicted values and experimental values.

5. Please correct any grammatical errors in the article.

Author Response

On behalf of my co-authors, we would like to thank you for your constructive and helpful comments and the opportunity to revise the manuscript entitled “Output-Only Time-Varying Modal Parameters Identification Method Based on TARMAX Model in Milling of Thin-Walled Workpiece” (Manuscript ID: micromachines-1888832).

Those comments are all valuable and very helpful to improve our manuscript, and provide some significant guidance for our further research. We have studied comments carefully and revised the manuscript. And, we responded point by point to reviewer’s comments as listed below. In addition, the relevant changes are highlighted in red in the attached version. We hope these will make the manuscript more acceptable for publication.

The responses to the reviewer’s comments are as following:

Comment 1: The method proposed in this paper is used to obtain the in-process modal parameters of thin-walled parts. However, there are only prediction data of natural frequency in the prediction result. Is it possible to supplement the predicted results for the magnitude of the frequency response function?

Response:

Thank you for your valuable suggestion.

Your comment is very helpful for improving our papers’ preciseness. We are very sorry to express the function of the frequency response function in this paper. As you known, the frequency response function is very important for predicting the stability in milling of thin-walled workpiece, and we have been actively looking for different methods to address this problem. However, in this paper, we focus on calculating the modal parameters through the in-process dynamic milling response in milling, then, several frequency response functions and its magnitude are obtained to validate the effectiveness of the calculated method under different machining stage. In addition, the unreasonable expressions have revised in the paper, and the changes have been highlighted in red in the revised manuscript.

Comment 2: Please confirm that the description of the cutting force in Eq. 15 is correct. Please explain why only the cutting forces in the x and y directions are applied and the cutting forces in the z direction are ignored.

Response:

Thank you for pointing it out.

Your comment is very helpful to improve our manuscript. We admit that this is a very obvious confusion in the description of the cutting force in Eq. 15, we are sorry for your misunderstanding. As is known to all, the machining efficiency of flank milling is very high, therefore, this paper mainly investigates the method of modal parameters in-process identification in flank milling. Considering workpiece clamping method and tool status, the natural frequencies in x and y direction is lower than that in z direction, and the main vibration modes is in x and y direction. In addition, with material removal, the dynamic stiffness of the workpiece changes obviously in x and y direction, which cause chatter in milling. Therefore, for simplification, the proposed method ignores the cutting forces in the z direction. And, a detailed explanation of it is provided in section 2.2.1in the paper. Then, the changes have been highlighted in red in the revised manuscript.

Comment 3: In the page 15, please confirm that the identified cutting force coefficients are correct. In general, the tangential shear force coefficient ktc should be larger than the radial shear force coefficient krc.

Response:

Thank you for pointing it out.

Your comment is very helpful to improve our papers’ preciseness. We confirm that the calibrated cutting force coefficients are correct, and the predicted cutting force agree well with the experimental results, which can be seen from Fig.1 under spindle speed 1000r/min, axial depth of  cut 0.5mm, feed speed 160mm/min.

Fig.1 Predicted and experimental milling forces

From Fig.1, it can be found that the predicted cutting force agree well with the experimental results, and the milling force in the x direction is larger than that in the y-direction, which cause that the tangential shear force coefficient ktc is less than the radial shear force coefficient krc. Then, the relevant modifications are made and highlighted in red in the revised manuscript.

Comment 4: The error analysis between the predicted results and the experimental results is carried out in the article. Please list the predicted values and experimental values.

Response:

Thank you for pointing it out.

Your comment is very helpful to improve our manuscript. We are sorry for your misunderstanding because we don’t clearly express the error analysis between the predicted results and the experimental results. In milling, the modal parameters cannot be obtained in real time, so it is assumed that the natural frequency changes linearly in continuous processing. Then, 1-order natural frequency predicted results and estimated results in different milling stages are obtained and shown in Fig.2.

From Fig.2, it is found that the predicted natural frequencies using the proposed method agree well with the experimental values, and the predicted values are slightly higher than the Experimental values due to the coupling benefits of the tool and the workpiece during machining, which is consistent with the experimental phenomena in literature 15. In addition, several scatter points that offset the baseline under machining start and end stage. The reason for this is that the milling is at the beginning or end of cut at this time, and the vibration displacement response obviously changes, which causes the large deviation between the predicted values and the experimental values.

Fig.2. 1-order natural frequency predicted results and estimated results in different milling stages. (line: experimental values; dot: estimated values).

   To further demonstrate the validity of the method, Mean Absolute Error (MAE), and Mean Relative Error (MRE) were introduced, calculated and shown in Fig.3.

Fig.3. Error between predicted results and experimental results. (black line: MAE; red line: MRE).

From Fig.3, the average absolute error of the TARMAX method is around 20Hz and the relative error is around 4.3%, and the prediction results are relatively stable. The third group of experiments had the highest prediction accuracy because a smoother dis-placement signal was chosen for the third group of experiments. There were some non-smooth signals in the displacement signals in the other sets of experiments. As the number of milling experiments increases, the relative error of the TARMAX method shows an upward trend. And the main reasons for this are as follows. As the number of experiments increases, the tool wears faster and dynamic milling forces change significantly, which can cause fluctuations in vibration response, which is consistent with the fourth and fifth experiments in which the displacement signals included some unsteady components. And, the changes have been highlighted in red in the revised manuscript.

Comment 5: Please correct any grammatical errors in the article.

Response:

Thank you for pointing it out.

Your comment is very helpful to improve our manuscript. We admit that this is a very obvious mistake. The entire manuscript has been read thoroughly and we have made our utmost efforts to make the presentation as professional as possible. And, the changes have been highlighted in red in the revised manuscript.

To sum up, all questions have been answered.

We appreciate for Editor/Reviewer’ warm work earnestly, and hope that the correction will meet with approval. Once again, thank you very much for your comments and suggestions.

Sincerely yours,

Junjin Ma

Reviewer 2 Report

In this paper, the TARMAX model considering external force excitation is constructed to characterize the actual situation of thin-walled workpiece milling and the corresponding simulation model is established. The method has high accuracy and effectiveness through experimental verification. This paper is rich in content, but it needs a major revision according to the following comments: 

1. The paper lacks innovation and there have been related papers published.

2. The advantages of TARMAX method and the reasons for choosing 3-DOF time-varying structure should be added.

3. The statement of “the relative error of the TARMAX method is gradually increased” needs more detailed analysis with experiment replications and error graph to verify.

4. The error analysis of experimental results is not deep enough.

5. In Fig.6, why only the signal-to-noise ratio of 15dB was chosen for further research?

6. The figure quality and the paper format should be enhanced, for example, Figures 7 and 8 are very blurry.

7. The following paper is a recently published review which discusses the thin-walled titanium alloys components and may be helpful for the introduction part of this paper:

[1] Bao Y, Wang B, He Z, Kang R, Guo J. Recent progress in flexible supporting technology for aerospace thin-walled parts: A review [J]. Chinese Journal of Aeronautics, 2021, 35: 10-26.

[2] Wang K H, Wang L L, Zheng K L, He Z B, Politis D J et al. High-efficiency forming processes for complex thin-walled titanium alloys components: state-of-the-art and perspectives [J]. International Journal of Extreme Manufacturing, 2020, 2: 032001.

[3] Guo J, Xu Y, Pan B, Zhang J, Kang R, Huang W, Du D. A New Method for Precision Measurement of Wall-Thickness of Thin-Walled Spherical Shell Parts[J]. Micromachines, 2021, 12: 467.

Author Response

Dear Editor and Reviewer,
On behalf of my co-authors, we would like to thank you for your constructive and helpful comments and the opportunity to revise the manuscript entitled “Output-Only Time-Varying Modal Parameters Identification Method Based on TARMAX Model in Milling of Thin-Walled Workpiece” (Manuscript ID: micromachines-1888832).
Those comments are all valuable and very helpful to improve our manuscript, and provide some significant guidance for our further research. We have studied comments carefully and revised the manuscript. And, we responded point by point to reviewer’s comments as listed below. In addition, the relevant changes are highlighted in red in the attached version. We hope these will make the manuscript more acceptable for publication.

The responses to the reviewer’s comments are as following:

In this paper, the TARMAX model considering external force excitation is constructed to characterize the actual situation of thin-walled workpiece milling and the corresponding simulation model is established. The method has high accuracy and effectiveness through experimental verification. This paper is rich in content, but it needs a major revision according to the following comments:
Comment 1: The paper lacks innovation and there have been related papers published.
Response:
Thank you for your valuable suggestion.
Your comment is very helpful to improve our manuscript. We are very sorry that the innovation of the paper does not express clearly. As you known, the TARMA method used to predict time-varying modal parameters have been investigated by many researchers in bridges, buildings, and aerocraft field et al. However, all of the above theories are based on the assumption that the system external excitation is environmental excitation. Nevertheless, our research mainly focuses on the on-line identification of modal parameters in milling of thin-walled workpiece, and the system external excitation of the strong harmonic signals and white noise signals are considered, which is closer to the actual milling conditions. Under these circumstances, the TARMA method cannot be used directly to predict time-varying modal parameters, therefore, to solve this problem, we propose a TARMAX model which treats the milling force and the white noise signal together as external excitation signals, which can accurately predict time-varying modal parameters, which can be seen form Fig.14 and Fig.15 in the paper. 
In addition, the unreasonable expressions have revised in the paper, and the changes have been highlighted in red in the revised manuscript.

Comment 2: The advantages of TARMAX method and the reasons for choosing 3-DOF time-varying structure should be added.
Response:
Thank you for pointing it out. 
Your comment is very helpful for improving our papers’ preciseness. We admit that this is a very obvious confusion in expressing the advantages of TARMAX method and the reasons for choosing 3-DOF time-varying structure. According to you suggestion, we have added the advantages of the TARMAX method in Section 2. In addition, the reason of choosing a 3-degree-of-freedom time-varying structure is elaborated in Section 2, which can make the manuscript more accurate and clear. Then, a detailed revision is provided in the paper, and the changes have been highlighted in red in the revised manuscript.

Comment 3: The statement of “the relative error of the TARMAX method is gradually increased” needs more detailed analysis with experiment replications and error graph to verify.
Response:
Thank you for pointing it out.
We take it. Your comment is very helpful to improve our papers’ preciseness. In the revised version, we have added a more detailed analysis for the relative error of the TARMAX method in Section 4.3, in addition, to verify the calculated accuracy, an error graphs are plotted and shown in Figure 15, which can make the experimental analysis part more comprehensive and rich. Finally, the relevant modifications are highlighted in red in the revised manuscript.

Comment 4: The error analysis of experimental results is not deep enough.
Response:
Thank you for pointing it out.
We take it. Your comment is very helpful to improve our manuscript. We are sorry for the less error analysis of experimental results. We have added a more comprehensive error analysis in Section 4.3 to make the error analysis part deep sufficiently. Finally, the relevant modifications are highlighted in red in the revised manuscript.

Comment 5: In Fig.6, why only the signal-to-noise ratio of 15dB was chosen for further research?
Response:
Thank you for pointing it out.
Your comment is very helpful to improve our papers’ preciseness. We admit that this is a very obvious confusion, we are sorry for your misunderstanding. In this paper, we have investigated the effect of signal-to-noise ratios of 30dB, 20dB, 15dB on the results, then, the original signal, the signal-to-noise ratio of 30dB, 20dB displacement signals are shown in Fig.1, Fig.2 and Fig.3, and the effect of signal-to-noise ratios of 15dB on the results (in Figure 6 in revised manuscript). In the process of writing the paper, for simplification, only the signal-to-noise ratio of 15dB was chosen for the paper, because a signal-to-noise ratio of 15dB has a greater impact on the system and some obvious noise interference can be found in the signals. Finally, the detailed modifications are highlighted in red in the revised manuscript. 

Fig. 1 Displacement response of a 3-DOF time-varying structure without noise contamination.

Fig. 2 Displacement response of 3-DOF time-varying structure with signal-to-noise ratio of 30.

Figure 3 Displacement response of 3-DOF time-varying structure with signal-to-noise ratio of 20.

Comment 6: The figure quality and the paper format should be enhanced, for example, Figures 7 and 8 are very blurry.
Response:
Thank you for pointing this out.
Your comment is very helpful to improve our manuscript. We are very sorry for the blurry in the paper. Then, we try our best to improve the pictures, especially, Figures 7 and 8. And, the changes have been highlighted in red in the revised manuscript.

Comment 5: The following paper is a recently published review which discusses the thin-walled titanium alloys components and may be helpful for the introduction part of this paper:
[1] Bao Y, Wang B, He Z, Kang R, Guo J. Recent progress in flexible supporting technology for aerospace thin-walled parts: A review [J]. Chinese Journal of Aeronautics, 2021, 35: 10-26.
[2] Wang K H, Wang L L, Zheng K L, He Z B, Politis D J et al. High-efficiency forming processes for complex thin-walled titanium alloys components: state-of-the-art and perspectives [J]. International Journal of Extreme Manufacturing, 2020, 2: 032001.
[3] Guo J, Xu Y, Pan B, Zhang J, Kang R, Huang W, Du D. A New Method for Precision Measurement of Wall-Thickness of Thin-Walled Spherical Shell Parts[J]. Micromachines, 2021, 12: 467.
Response:
Thank you for pointing this out.
Your comment is very helpful to improve our manuscript. We take it. As you suggested, these references can be helpful for the introduction part of this paper. Therefore, the three papers are added to the introduction, which is shown in References section. And, the changes have been highlighted in red in the revised manuscript.

To sum up, all questions have been answered.

We appreciate for Editor/Reviewer’ warm work earnestly, and hope that the correction will meet with approval. Once again, thank you very much for your comments and suggestions.

Sincerely yours,
Junjin Ma

Reviewer 3 Report

This article considered the identification of time-varying modal parameters of thin-walled parts under milling force excitation. The relationship between the modal parameters and the time-varying coefficients of the TARMAX model is established, and the time-varying coefficients are estimated by the least squares method. However, improvements to the present version are necessary, as given below.

1. Some grammatical and word errors need to be corrected.

2. Some of the matrices in section 2 need to be bolded.

3. Were the MAE values in Sections 3 and 4 solved using equation 54?

4. The 16s displacement data in section 4 needs to be labelled in Figure 13.

Author Response

On behalf of my co-authors, we would like to thank you for your constructive and helpful comments and the opportunity to revise the manuscript entitled “Output-Only Time-Varying Modal Parameters Identification Method Based on TARMAX Model in Milling of Thin-Walled Workpiece” (Manuscript ID: micromachines-1888832).

Those comments are all valuable and very helpful to improve our manuscript, and provide some significant guidance for our further research. We have studied comments carefully and revised the manuscript. And, we responded point by point to reviewer’s comments as listed below. In addition, the relevant changes are highlighted in red in the attached version. We hope these will make the manuscript more acceptable for publication.

The responses to the reviewer’s comments are as following:

This article considered the identification of time-varying modal parameters of thin-walled parts under milling force excitation. The relationship between the modal parameters and the time-varying coefficients of the TARMAX model is established, and the time-varying coefficients are estimated by the least squares method. However, improvements to the present version are necessary, as given below.

Comment 1: Some grammatical and word errors need to be corrected.

Response:

Thank you for your valuable suggestion.

Your comment is very helpful to improve our manuscript. We have been read the manuscript thoroughly and made our utmost efforts to make the presentation as professional as possible. In addition, we entrust the professional press to do proof-reading. Finally, the relevant detailed changes are highlighted in red in the attached version.

Comment 2: Some of the matrices in section 2 need to be bolded.

Response:

Thank you for pointing it out.

Your comment is very helpful for improving our papers’ preciseness. We have bolded the matrices in the article and in addition, corrected errors in the presentation of matrices in the article. And, the relevant detailed changes are highlighted in red in the attached version.

Comment 3: Were the MAE values in Sections 3 and 4 solved using equation 54?

Response:

Thank you for pointing it out.

We take it. Your comment is very helpful to improve our papers’ preciseness. We apologize that we did not express it clearly. Therefore, we added the solved method in the paper in Section 4.3. Then, the MAE in Section 3 is calculated using Equation 54, and the MAE in Section 4 is solved using Equation 59. In addition, we have added the solution to the MRE in the text. Finally, the relevant detailed changes are highlighted in red in the attached version.

Comment 4: The 16s displacement data in section 4 needs to be labeled in Figure 13.

Response:

Thank you for pointing it out.

We take it. Your comment is very helpful to improve our manuscript. We have added the corresponding content to the article in Section 4.3 and have marked it in Figure 13.

To sum up, all questions have been answered.

We appreciate for Editor/Reviewer’ warm work earnestly, and hope that the correction will meet with approval. Once again, thank you very much for your comments and suggestions.

Sincerely yours,

Junjin Ma

Reviewer 4 Report

This manuscript focuses on a TARMAX model considering milling force excitation for identifying the dynamic modal parameters. In this paper, a TARMAX model is proposed, and numerical simulations and machining tests were conducted to verify the effectiveness of the proposed method. In general, this manuscript has organized well and written well. Some important results have been obtained. Before considering accepting this paper, some little problems should be solved.

 1. The matrices in Equations 14 and 15 need to be denoted by "[ ]".

2. In the page 7, should be changed to.

3. There are many grammatical errors in the article that need to be corrected.

Author Response

On behalf of my co-authors, we would like to thank you for your constructive and helpful comments and the opportunity to revise the manuscript entitled “Output-Only Time-Varying Modal Parameters Identification Method Based on TARMAX Model in Milling of Thin-Walled Workpiece” (Manuscript ID: micromachines-1888832).

Those comments are all valuable and very helpful to improve our manuscript, and provide some significant guidance for our further research. We have studied comments carefully and revised the manuscript. And, we responded point by point to reviewer’s comments as listed below. In addition, the relevant changes are highlighted in red in the attached version. We hope these will make the manuscript more acceptable for publication.

The responses to the reviewer’s comments are as following:

This manuscript focuses on a TARMAX model considering milling force excitation for identifying the dynamic modal parameters. In this paper, a TARMAX model is proposed, and numerical simulations and machining tests were conducted to verify the effectiveness of the proposed method. In general, this manuscript has organized well and written well. Some important results have been obtained. Before considering accepting this paper, some little problems should be solved.

Comment 1: The matrices in Equations 14 and 15 need to be denoted by "[ ]".

Response:

Thank you for your valuable suggestion.

We take it. Your comment is very helpful to improve our manuscript. We have revised problem in the revised manuscript. And, the relevant detailed changes are highlighted in red in the attached version.

Comment 2: In the page 7, should be changed to.

Response:

Thank you for your valuable suggestion.

Your comment is very helpful to improve our manuscript. We have been read this section and correct the error. Finally, the relevant detailed changes are highlighted in red in the attached version.

Comment 3: There are many grammatical errors in the article that need to be corrected.

Response:

Thank you for pointing it out.

We take it. Your comment is very helpful to improve our papers’ preciseness. We apologize that we did not express it clearly. We have been read the manuscript thoroughly and made our utmost efforts to make the presentation as professional as possible. In addition, we entrust the professional press to do proof-reading. Finally, the relevant detailed changes are highlighted in red in the attached version.

To sum up, all questions have been answered.

We appreciate for Editor/Reviewer’ warm work earnestly, and hope that the correction will meet with approval. Once again, thank you very much for your comments and suggestions.

Sincerely yours,

Junjin Ma

Round 2

Reviewer 2 Report

In this paper, the tarmax model considering external force excitation is constructed to characterize the actual situation of thin-walled workpiece milling and the corresponding simulation model is established. The method has high accuracy and effectiveness through experimental verification. The quality of the article has been greatly improved after modification, but the following issues still need to be slightly adjusted and modified:

1. This research is very practical, but the description of the practical significance and application scenarios of this research needs to be strengthened in the introduction.

2. Is the reason for the increase of relative error mentioned in the conclusion of 4.3 determined to be caused by tool wear? Whether it has been verified by experiments?

3. The analysis of the results in 4.3 is not accurate enough. For example, how can we get the test value from the image that is slightly higher than the experimental value?

4. Figure 12 needs to be optimized. In addition, there are some errors in the writing of the article, please check carefully.

Author Response

On behalf of my co-authors, we would like to thank you for your constructive and helpful comments and the opportunity to revise the manuscript entitled “Output-Only Time-Varying Modal Parameters Identification Method Based on TARMAX Model in Milling of Thin-Walled Workpiece” (Manuscript ID: micromachines-1888832).

Those comments are all valuable and very helpful to improve our manuscript, and provide some significant guidance for our further research. We have studied comments carefully and revised the manuscript. And, we responded point by point to reviewer’s comments as listed below. In addition, the relevant changes are highlighted in red in the attached version. We hope these will make the manuscript more acceptable for publication.

The responses to the reviewer’s comments are as following:

In this paper, the tarmax model considering external force excitation is constructed to characterize the actual situation of thin-walled workpiece milling and the corresponding simulation model is established. The method has high accuracy and effectiveness through experimental verification. The quality of the article has been greatly improved after modification, but the following issues still need to be slightly adjusted and modified:

Comment 1: This research is very practical, but the description of the practical significance and application scenarios of this research needs to be strengthened in the introduction.

Response:

Thank you for your valuable suggestion.

Your comment is very helpful to improve our manuscript. We take it. In this paper, we have added the description of the practical significance and application scenarios of this research, which made the introduction more comprehensive. Then, a detailed revision is provided in the paper, and the changes have been highlighted in red in the revised manuscript.

Comment 2: Is the reason for the increase of relative error mentioned in the conclusion of 4.3 determined to be caused by tool wear? Whether it has been verified by experiments?

Response:

Thank you for pointing it out.

Your comment is very helpful for improving our papers’ preciseness. You know that tool wear have obvious effect on the dynamic cutting force verified by the reference [Cutting force model of longitudinal-torsional ultrasonic-assisted milling Ti-6Al-4V based on tool flank wear]. In addition, we also have conducted and observed the same phenomenon (spindle speed of 3184 r/min, axial depth of cut and radial depth of cut of 0.5 mm, feed rate 160 mm/min, down milling, four teeth milling cutter with a diameter of 6, workpiece material and dimensions shown in the manuscript). From experimental results, the average milling forces in the x and y directions increased from 8.63 N to 9.93 N and from 7.25 N to 8.05 N for the 1st and 14th experiments respectively, and tool wear was 0 mm and 0.06 mm respectively, which is shown that the milling force changes as the machining experiment progress (tool wear), and the relative error in prediction model increased during milling. In the future, we will continue to explore the prediction of modal parameters under consideration of tool wear in subsequent studies. Finally, the relevant modifications are highlighted in red in the revised manuscript.

Comment 3: The analysis of the results in 4.3 is not accurate enough. For example, how can we get the test value from the image that is slightly higher than the experimental value?

Response:

Thank you for pointing it out.

We take it. Your comment is very helpful to improve our papers’ preciseness. We admit that this is a very obvious confusion, we are sorry for your misunderstanding. In this paper, we have been read the manuscript thoroughly and made our utmost efforts to make the presentation as professional as possible. In addition, we entrust the professional press to do proof-reading. Finally, the relevant detailed changes are highlighted in red in the attached version.

Comment 4: Figure 12 needs to be optimized. In addition, there are some errors in the writing of the article, please check carefully.

Response:

Thank you for your valuable suggestion.

Your comment is very helpful to improve our manuscript. In the paper, we have optimized Figure 12, and we also revised other problems and made our paper more correctly, such as Figure 1.

In addition, the unreasonable expressions have revised in the paper, and finally, the changes have been highlighted in red in the revised manuscript.

To sum up, all questions have been answered.

We appreciate for Editor/Reviewer’ warm work earnestly, and hope that the correction will meet with approval. Once again, thank you very much for your comments and suggestions.

Sincerely yours,

Junjin Ma
